# Influence of Lifestyle Habits on Psychological Well-Being of University Students: A Quantitative Cross-Sectional Study

**DOI:** 10.3390/healthcare13101197

**Published:** 2025-05-20

**Authors:** Laura García-Pérez, Rosario Padial-Ruz, Mar Cepero-González, José Luis Ubago-Jiménez

**Affiliations:** Department of Didactics of Corporal Expression, Faculty of Education, University of Granada, 18071 Granada, Spain; mcepero@ugr.es (M.C.-G.); jlubago@ugr.es (J.L.U.-J.)

**Keywords:** physical activity, Mediterranean diet, sleep duration, resilience, self-esteem, psychological distress, mental health, psychological well-being, future teachers, university students

## Abstract

**Background**: This study explored the influence of three key lifestyle habits—physical activity (PA), adherence to the Mediterranean diet (MD), and sleep duration—on psychological well-being indicators (resilience, psychological distress, and self-esteem) in university students. **Methods**: A total of 928 students (67.3% females; total sample mean age = 21.01 ± 1.95) from the Faculty of Education at the University of Granada participated. Validated self-report instruments were used to assess all variables: the International Physical Activity Questionnaire (IPAQ-SF), the KIDMED index, the Connor–Davidson Resilience Scale (CD-RISC), the Depression Anxiety Stress Scales (DASS-21), and the Rosenberg Self-Esteem Scale. **Results**: Results showed that male students reported higher levels of PA, better MD adherence, longer sleep duration, and more favorable psychological outcomes. Stepwise regression analyses indicated that MD adherence was the strongest and most consistent predictor of resilience, self-esteem, and psychological distress. Sleep duration emerged as a key factor, particularly in predicting resilience among men. Although no direct association was found between PA and psychological distress, mediation analysis revealed a significant indirect effect through sleep duration. **Conclusions**: These findings underscore the relevance of promoting healthy lifestyle habits in an integrated and sex-sensitive manner to enhance mental health (MH) in university students. In particular, targeting future teachers may be essential, given their potential role as promoters of well-being in school settings.

## 1. Introduction

In recent years, mental health (MH) disorders among university students have increased significantly, further exacerbated by the impact of the COVID-19 pandemic [1,2,3,4]. Although some longitudinal studies have reported partial recoveries of post-pandemic symptoms such as depression [5] and stress [6], as well as increases in life satisfaction [7] and psychological resilience [8], most current research continues to report high levels of anxiety, stress, and depression in this population.

This scenario aligns with reports from international organizations, which emphasize the urgent need to strengthen MH systems [9]. In Europe, 16.98% of young people experience some type of mental disorder [10]. In Spain, the data follow a similar trend. The most comprehensive evaluation on this topic revealed that one in two students experiences moderate and/or severe levels of anxiety, and more than half have sought psychological help [11]. Moreover, a high prevalence of depressive symptoms (11.5–18.4%), anxiety (15.24–23.6%), and stress (34.5%) has been reported in Spanish universities [12,13]. In Mediterranean regions, one in three students has experienced a mental disorder in the past year, with severe functional impairment in one-third of these cases [14].

In this context, it is essential to examine personal variables associated with psychological well-being, such as self-esteem and resilience, which are widely recognized as protective factors against psychological distress [15,16]. High self-esteem has been linked to better academic adjustment [17], lower psychological distress [18,19], and a more positive self-perception [20]. Resilience, in turn, is negatively associated with anxiety and stress, and positively associated with indicators of positive mental health [21,22], positioning it as a key resource in demanding academic settings.

Along these lines, healthy lifestyle habits such as physical activity (PA), a balanced diet, and sleep have emerged as crucial factors in promoting psychological well-being and strengthening these personal resources.

In addition to preventing non-communicable diseases such as cardiovascular conditions [23], obesity [24], type 2 diabetes [25], or musculoskeletal disorders [26], PA plays a vital role in MH by reducing symptoms of stress, anxiety, depression, and suicidal ideation risk [27]. PA also contributes to increased self-esteem [28], improved mood [29], enhanced quality of life [30], better sleep [31], and greater resilience [32]. These benefits are not only individual but also social and economic, as they help reduce healthcare costs [33] and, specifically in educational contexts, positively impact academic performance [34].

Another key component is adherence to the Mediterranean diet (MD), regarded as one of the healthiest dietary patterns due to its benefits in preventing and treating chronic diseases [35,36,37]. Beyond its physiological impact, MD has been associated with greater mental well-being, including lower levels of psychological distress [38,39], and enhanced emotional and cognitive quality of life [40,41]. In addition to its protective role against distress, several studies highlight its positive relationship with personal resources such as self-esteem and resilience [41,42,43], fostering a positive self-image and greater academic and life coping capacity [44,45].

This life stage, marked by economic independence, academic stress, and reorganized routines, represents for many students the beginning of autonomous food management [46]. Consequently, low adherence to MD has been observed among Spanish university students, particularly during the first academic year, with rates ranging from 55% to 65% [47,48], and a downward trend as the university journey progresses [49,50].

Sleep also plays an essential role in MH and emotional balance [51]. Short or poor-quality sleep has been associated with higher levels of psychological distress [52,53,54], and insomnia may mediate the relationship between daily stressors and the emergence of emotional symptoms, amplifying their impact [55]. Additionally, sleep and resilience maintain a bidirectional relationship: while adequate rest promotes neurocognitive functioning and stress adaptation, insomnia weakens these capacities, increasing psychological vulnerability [56,57]. Moreover, good sleep quality is linked to greater emotional stability and a more positive self-perception, thereby enhancing self-esteem [58,59].

Despite the extensive evidence on the positive impact of these habits on MH, few studies have jointly analyzed their influence on variables such as self-esteem, resilience, and psychological distress. The potential benefits of these habits can be understood through several theoretical lenses. The biopsychosocial model emphasizes the dynamic interaction of biological (e.g., sleep), psychological (e.g., self-esteem and resilience), and social (e.g., academic and peer context) factors in shaping well-being [60]. Moreover, the Conservation of Resources theory conceptualizes self-esteem and resilience as essential psychological resources that protect individuals from MH problems [61], and these may be enhanced through healthy behaviors. In addition, the Theory of Planned Behavior suggests that students’ engagement in such habits may depend on attitudes, social norms, and perceived control [62], which in turn affect their MH.

Therefore, the aim of this study was to explore the influence of PA, adherence to the MD, and sleep duration on three key psychological variables—self-esteem, resilience, and psychological distress—in university students. The study is original in jointly addressing these habits and their role as protective factors for future educators, a population with great potential to influence child and adolescent development. Prior studies have documented sex-based differences in health behaviors and psychological well-being in university students, warranting stratified analysis by sex. It was hypothesized that higher levels of PA, greater adherence to the MD, and longer sleep duration would be associated with higher self-esteem and resilience, and lower psychological distress. It was also expected that these associations would differ by sex.

## 2. Materials and Methods

### 2.1. Study Design

A cross-sectional, non-experimental, and analytical study design was employed, with a retrospective approach and a descriptive–comparative and correlational component. The study was conducted between February and June 2024.

### 2.2. Participants

The participants were university students aged between 18 and 26 years, enrolled in the Faculty of Education Sciences at the University of Granada, Spain. This faculty offers four degree programs, among which Early Childhood Education and Primary Education degrees were specifically selected for strategic reasons related to logistical feasibility and the high number of enrolled students. According to official enrollment data from the 2023/2024 academic year, there were 3363 students registered in education-related degree programs (Early Childhood Education and Primary Education) at the University of Granada. Therefore, the final sample is approximately 27.6% of the total population within this faculty. Although the sampling method was non-random and based on voluntary participation, the large sample size and inclusion of students from different academic years contribute to the robustness and contextual validity of the findings.

Inclusion criteria were (a) being actively involved in Early Childhood or Primary Education degree programs at the University of Granada; (b) being between 18 and 26 years old; and (c) providing informed consent to voluntarily participate in the study.

Exclusion criteria included (a) incomplete or inconsistent responses in the questionnaire; and (b) current use of medication for sleep regulation or psychological disorders, due to their potential impact on the study variables.

Initially, 1008 students participated, but 80 were excluded based on these criteria. Therefore, the final analysis included 928 students (625 females and 303 males), with a mean age of 21.01 ± 1.95 years.

### 2.3. Instruments and Variables

#### 2.3.1. IPAQ-SF

The short version of the International Physical Activity Questionnaire (IPAQ-SF) was used to assess overall PA levels [63]. This questionnaire has demonstrated reliability coefficients above 0.65 and a pooled correlation of ρ = 0.76 (95% CI: 0.73–0.77), which has been confirmed in studies involving university students [64]. It consists of 7 items assessing the frequency and duration of PA in 3 intensity categories: light (3.3 METs), moderate (4.0 METs), and vigorous (8.0 METs). PA levels were classified using the criteria established by the IPAQ Committee [65]. High PA level corresponds to individuals accumulating at least 3000 MET-min/week of combined activity. Moderate level includes participants engaging in at least 600 MET-min/week. Low level applies to those who do not meet the above criteria, indicating low or no PA. The total PA score was calculated by summing the MET values for all reported activities (days × min × METs), resulting in a combined PA score expressed in MET-min/week [64].

#### 2.3.2. KIDMED

The updated Spanish version of the Mediterranean Diet Quality Index (KIDMED) questionnaire was used to assess adherence to the MD [66]. This questionnaire consists of 16 dichotomous items, 4 of which are negatively worded and 12 positively worded. The total score ranges from −4 to 12. Based on this score, adherence is classified into three levels: poor (≤3), needs improvement (4–7), and optimal (≥8). The KIDMED has been previously used in similar research [67] and has demonstrated high reliability and validity, with excellent internal consistency (α = 0.88; 95% CI: 0.87–0.89).

#### 2.3.3. CD-RISC

To assess resilience, the Spanish version of the Connor–Davidson Resilience Scale (CD-RISC 25) validated by Manzano-García and Ayala-Calvo was used [68]. This scale has shown strong validity and reliability in university student populations [69,70,71].

It consists of 25 items rated on a 5-point Likert scale (0 = not true at all to 4 = almost always true), yielding a total score between 0 and 100, where higher scores indicate greater resilience. In the present study, the scale demonstrated excellent internal consistency (α = 0.84; 95% CI: 0.82–0.85).

#### 2.3.4. DASS-21

The DASS-21 scale was used to evaluate psychological distress, measuring three key MH dimensions: depression, anxiety, and stress [72]. The Spanish version validated by Daza et al. was applied [73], which has demonstrated high reliability and validity, with excellent internal consistency (α = 0.94; 95% CI: 0.94–0.95), and has been widely used in research with both Spanish [74] and international students [75,76]. It consists of 21 items rated on a 4-point Likert scale from 0 (not at all applicable) to 3 (very applicable). A global psychological distress score was computed by summing all items, with a total subscale score ranging from 0 to 21.

#### 2.3.5. Rosenberg Self-Esteem Scale

Self-esteem was assessed using a Spanish-adapted and validated version of the Rosenberg Self-Esteem Scale [77,78], previously used in university student populations [17]. The scale showed high internal consistency (α = 0.84; 95% CI: 0.80–0.87), confirming its validity and reliability. It comprises 10 items, 5 worded positively (items 1, 3, 4, 7, 10) and 5 negatively (items 2, 5, 6, 8, 9), designed to control response bias. Items are scored on a four-point scale (A–D), from 1 = strongly disagree to 4 = strongly agree. The total score ranges from 10 to 40, with higher scores indicating greater self-esteem.

#### 2.3.6. Average Hours of Sleep

Self-reported questions regarding sleep habits were used to assess average sleep duration. Participants reported their average sleep time during weekdays and weekends. These two values were then combined to calculate a representative weekly average sleep duration.

Sleep duration was categorized into two groups using a 7 h cutoff point. This decision was based on scientific and empirical recommendations. The National Sleep Foundation indicates that young adults typically require 7 to 9 h of sleep per day to maintain adequate health [79]. Chaput et al., after analyzing 41 studies across 14 countries, concluded that sleeping around 7–8 h per day is associated with better physical and mental health indicators [80]. Additionally, sleeping less than 7 h regularly has been linked to significantly increased physical and psychological distress [81]. For these reasons, the 7 h threshold was adopted in this study.

### 2.4. Procedure

Data collection was conducted online between February and June 2024. A convenience sampling approach was used. Students were recruited through multiple coordinated channels, including official university email distribution lists and in-class announcements with the collaboration of faculty members. Participation was voluntary and anonymous. Prior to completing the questionnaire, participants were informed about the study’s objectives, procedures, and data confidentiality, and they provided informed consent in accordance with ethical guidelines. The questionnaire took approximately 12 min to complete. As an incentive, participants were entered into a raffle for a prize redeemable for computer and tech-related equipment.

### 2.5. Ethical Considerations

The study was conducted in accordance with the guidelines of the Declaration of Helsinki (1964) and was approved by the Ethics Committee of the University of Granada (3678/CEIH/2023). All data were processed in compliance with European data protection legislation, specifically Regulation (EU) 2016/679 (GDPR) of the European Parliament and Council of 27 April 2016.

### 2.6. Statistical Analysis

A post hoc power analysis was conducted using G*Power 3.1.9.7 to assess the adequacy of the sample size. Considering an alpha level of 0.05, three predictors (physical activity, Mediterranean diet, and sleep duration), and an expected medium effect size (f^2^ = 0.15), the analysis indicated that a sample of 928 participants provides a statistical power of 1.00. These results confirm that the sample size was sufficient to detect meaningful associations in the regression analyses. A descriptive analysis of sociodemographic variables was conducted using measures of central tendency and dispersion (mean and standard deviation for quantitative variables, frequencies, and percentages for categorical variables). The Kolmogorov–Smirnov test with Lilliefors correction was used to assess data normality. Since the data were not normally distributed, Mann–Whitney and Kruskal–Wallis tests were used for comparisons of continuous variables, and the Chi-square test was used for categorical variables.

To explore the relationships between quantitative variables, Spearman’s rank correlation was performed, controlling for sex. Finally, multiple regression analysis was carried out separately by sex. A *p*-value of less than 0.05 was considered statistically significant for all analyses. Statistical analyses were performed using IBM SPSS Statistics^®^ for Windows, Version 28.0, Armonk, NY, USA: IBM Corp USA.

## 3. Results

Table 1 presents the levels of combined PA, adherence to the MD, sleep duration, psychological resilience, psychological distress, and self-esteem, disaggregated by sex.

Significant sex differences were found across all analyzed variables. Male students reported higher scores in all healthy lifestyle indicators, including greater levels of combined PA (5016.20 ± 2303.98 vs. 3844.56 ± 2170.89; *p* < 0.001), higher adherence to the MD (7.57 ± 0.63 vs. 7.30 ± 0.65; *p* < 0.001), and longer sleep duration (7.57 ± 0.63 vs. 7.30 ± 0.65; *p* < 0.001).

Similarly, male students showed better results in the psychological variables assessed compared to female students. Specifically, they reported higher resilience to adverse situations (65.59 ± 6.75 vs. 63.24 ± 7.89; *p* < 0.001), greater self-esteem (32.31 ± 4.78 vs. 29.20 ± 5.19; *p* < 0.001), and lower levels of psychological distress (35.50 ± 22.61 vs. 49.75 ± 27.39; *p* < 0.001).

Table 2 presents the characteristics of the sample according to levels of combined PA, adherence to the MD, and reported sleep duration.

Firstly, differences are shown based on the level of combined PA. Significant differences were found in all variables assessed, except for psychological distress. Students with a higher level of PA reported more hours of sleep compared to those with moderate PA (7.55 ± 0.66 vs. 7.14 ± 0.61; *p* < 0.001) and low PA (7.55 ± 0.66 vs. 7.08 ± 0.52; *p* < 0.001). No significant differences were found between the low and moderate PA groups. Regarding adherence to the MD, significant differences were observed between the low and high PA groups (*p* < 0.001), and between the moderate and high PA groups (*p* < 0.001). However, no significant differences were found between the low and moderate groups. In this sense, participants with higher PA levels reported better adherence to the MD (7.24 ± 2.03) than those with moderate (5.77 ± 2.23) or low (5.17 ± 2.31) PA levels. In terms of resilience and self-esteem, significant differences were found only between the moderate and high PA groups, with the high PA group showing greater resilience (64.77 ± 7.10 vs. 62.63 ± 8.19, *p* < 0.001) and higher self-esteem (30.60 ± 5.22 vs. 29.40 ± 5.20; *p* = 0.008).

Secondly, differences are presented according to the level of adherence to the MD. Significant differences were observed across all variables analyzed. Students with optimal adherence to the MD reported higher levels of PA (5284.71 ± 2200.68) compared to those with moderate (3611.41 ± 2040.50) and low adherence or poor diet quality (2766.11 ± 1837.44), with significant differences between all three levels (*p* < 0.001). Additionally, the optimal adherence group reported more hours of sleep (7.64 ± 0.63) and greater coping capacity (67.05 ± 5.41). However, no significant differences were found between the low and moderate adherence groups. The optimal adherence group also showed lower psychological distress and higher self-esteem compared to the low (*p* < 0.001) and moderate (*p* < 0.001) adherence groups.

Lastly, students who reported sleeping less than 7 h engaged in less PA (3415.12 ± 2101.68 vs. 4415.82 ± 2281.18; *p* < 0.001) and had lower diet quality (5.76 ± 2.21 vs. 6.91 ± 2.17; *p* < 0.001). They also exhibited higher levels of psychological distress (60.02 ± 24.89 vs. 41.63 ± 25.99; *p* < 0.001), lower resilience (61.32 ± 7.98 vs. 64.64 ± 7.39; *p* < 0.001), and lower self-esteem (27.47 ± 4.68 vs. 30.81 ± 5.18; *p* < 0.001).

Table 3 displays the correlation coefficients between lifestyle habits (PA, adherence to the MD, and sleep duration) and the psychological factors assessed (psychological distress, resilience, and self-esteem).

A longer duration of combined physical activity showed its strongest and most positive correlation with adherence to the MD (r = 0.445; *p* = 0.000). It also showed moderate positive correlations with sleep duration (r = 0.327; *p* = 0.000), resilience (r = 0.182; *p* = 0.005), and self-esteem (r = 0.164; *p* = 0.000).

Sleep duration showed a strong positive correlation with self-esteem (r = 0.506; *p* = 0.000), as well as moderate positive correlations with resilience (r = 0.447; *p* = 0.000) and adherence to the MD (r = 0.352; *p* = 0.000). In contrast, it was moderately and negatively correlated with psychological distress (r = −0.417; *p* = 0.000).

To fulfill the objective of this study, a stepwise regression analysis was conducted for each dependent variable (resilience, psychological distress, and self-esteem), disaggregated by sex (see Table 4 and Table 5). Additionally, multicollinearity was assessed using collinearity statistics. All models were statistically significant and showed a stronger impact among male university students.

First, in the resilience model, the explained variance was higher in males (34.3%; F = 52.095, *p* < 0.001) than in females (21.6%; F = 56.971, *p* < 0.001), indicating that the predictors included in the model had a greater influence on resilience among male students, although both models were significant. For males, the strongest predictor was sleep duration (β = 0.475, *p* < 0.001), whereas for females, the most influential factor was adherence to the MD (β = 0.352, *p* < 0.001). Physical activity showed a positive effect on resilience, but only in males (β = 0.133, *p* = 0.014).

Second, in the psychological distress model, the explained variance was again higher in males (48.1%; F = 92.498; *p* < 0.001) than in females (41.4%; F = 146.523; *p* < 0.001). This suggests that factors such as diet, sleep, and physical activity account for nearly half of the variability in psychological distress in this population, with slightly greater influence among males. Adherence to the MD was the strongest predictor in both sexes, with a greater impact in males (β = −0.640, *p* < 0.001 vs. β = −0.562, *p* < 0.001), indicating that poor diet quality is associated with higher levels of psychological distress. Sleep duration was also a key factor in both models, with similar effects in both groups (β = −0.291 vs. β = −0.294; *p* < 0.001). Combined physical activity showed a positive association with psychological distress, with a more pronounced effect among males (β = 0.419, *p* < 0.001 vs. β = 0.294, *p* < 0.001).

Lastly, in the self-esteem model, the explained variance was greater among male students (52.4%; F = 109.626; *p* < 0.001) than among females (43%; F = 156.217; *p* < 0.001). In both sexes, adherence to the MD was the strongest predictor of self-esteem (β = 0.547, *p* < 0.001 vs. β = 0.541, *p* < 0.001), with a similar effect in both groups. Sleep duration also showed a significant contribution to the model, suggesting a stronger influence on male students’ self-esteem (β = 0.422, *p* < 0.001 vs. β = 0.331, *p* < 0.001). In contrast, combined physical activity was negatively associated with self-esteem in both sexes (β = −0.232, *p* < 0.001 for males and β = −0.186, *p* < 0.001 for females).

### Exploratory Mediation Analysis: The Role of Sleep

After observing no significant relationship between PA level and psychological distress in the comparative analyses (*p* = 0.283), an exploratory mediation analysis was conducted to examine whether this relationship could occur indirectly through sleep duration. This decision was based on previous literature, which has reported negative associations between PA and psychological distress [82,83], as well as positive associations between PA and sleep indicators [84].

A simple mediation model (Model 4 of PROCESS v4.0 for SPSS) was used [85], in which PA was included as the predictor variable (X), psychological distress as the dependent variable (Y), and average sleep duration as the mediating variable (M), controlling for participant sex.

The analysis revealed a significant negative indirect effect through sleep (B = −0.014; 95% CI: [−0.018, −0.011]; *p* < 0.001), suggesting that part of the effect of PA on psychological distress occurs via increased sleep duration.

This finding highlights a complex dynamic, in which the protective effect of PA may not be direct but rather mediated by the improvement of other healthy habits such as nighttime rest. The model explained 21.1% of the total variance in psychological distress (R^2^ = 0.211; F(3,924) = 81.80; *p* < 0.001). Figure 1 illustrates the proposed mediation model and its corresponding coefficients.

## 4. Discussion

The aim of the present study was to explore the influence of three lifestyle habits—PA, sleep duration, and adherence to the MD—on key psychological well-being variables, namely, resilience, psychological distress, and self-esteem, in university students. Despite the extensive literature supporting the individual benefits of these habits on MH, studies that examine their joint impact, especially in university populations, remain limited. This study aimed to bridge this gap by not only examining the combined influence of PA, sleep, and MD adherence on MH, but also by exploring sex-based differences and identifying the most salient lifestyle predictors of each psychological outcome in a sample of future educators. Although the initial hypothesis posited a direct association between PA and psychological distress, the absence of a significant direct effect led us to explore an indirect pathway through sleep duration, based on prior evidence of such associations.

Overall, the results showed that male students presented better indicators in both lifestyle habits and psychological well-being variables, suggesting a sex-based differential pattern. Firstly, males showed significantly higher levels of PA, which is consistent with previous studies of a similar nature [86]. This difference may be explained by the fact that male university students, from an early age, tend to show more self-determined motivation toward PA, associated with competition, enjoyment, and immediate reward. In contrast, females tend to exhibit more extrinsic motivation, possibly linked to long-term goals, which may hinder adherence to PA [87].

Secondly, regarding sleep, males reported longer nighttime sleep duration. This discrepancy may be due to findings indicating that females are at greater risk of experiencing poorer sleep quality and spend less time resting, which is also associated with a worse perception of health status [88].

Thirdly, the results showed that males reported better diet quality than females. This finding contrasts with most previous studies, which suggest that females usually show greater adherence to healthy dietary patterns [88,89,90]. This difference has been partially attributed to greater concern for self-care, weight control, and body image [91,92]. However, the evidence is not conclusive, as some studies indicate that males can equal or even surpass females in dietary quality [93,94,95] or find no significant sex differences [96,97]. These inconsistencies may be related to the interaction of personal and contextual factors such as education level, living environment, sleep quality, PA, or health status, which were notably altered during the COVID-19 pandemic [98].

Males also showed more favorable scores in resilience and self-esteem, and lower levels of psychological distress compared to females, consistent with prior research documenting significant sex differences in psychological variables. Several studies indicate that males tend to report higher levels of resilience [99,100,101]. It has been observed that males aged 20–29 years show significantly greater resilience than females in the same age group, possibly due to sociocultural influences of gender roles, which reinforce self-efficacy and active coping strategies in males [102,103]. However, in later stages, females report higher or at least similar resilience levels as they reach adulthood [104,105]. Thus, the initial male advantage tends to diminish with age, reinforcing the conception of resilience as a dynamic and evolving process shaped by life experiences rather than a fixed gender trait.

Similarly, some studies report higher self-esteem levels in males [106,107], possibly related to a lower expression of vulnerability [108] and a more positive self-image supported by greater personal control [109]. One possible explanation is males’ greater tendency to exaggerate desirable physical traits, which may enhance their self-esteem and body satisfaction [110]. However, females are more likely to experience higher levels of depression, anxiety, and stress, particularly in high-demand academic contexts [111]. This symptomatology could be related to greater sensitivity to the social environment [112], a lower self-perception of progress in leadership roles [113], higher emotional empathy [114], or broader life concerns [115], which may account for the differences observed in this study.

Descriptive results revealed a consistent pattern supporting the positive influence of healthy lifestyle habits on students’ psychological well-being. Specifically, participants with higher levels of PA, greater adherence to the MD, and longer nighttime sleep duration presented a more favorable psychological profile, characterized by higher resilience and self-esteem. However, no significant differences were found in psychological distress according to PA level, suggesting that this behavior may not directly influence its reduction. Several studies have indicated that while PA offers recognized psychological benefits [82,116], its relationship with symptoms of stress, anxiety, or depression is not always direct, particularly in non-clinical populations [117]. In this regard, the literature suggests that PA’s effect on MH may be modulated by various factors such as intrinsic motivation, the type of exercise performed, perceived competence, or even the quality of the environment in which it takes place [31,118].

Moreover, although PA was not significantly associated with psychological distress in descriptive and comparative analyses, it was positively correlated with sleep duration, which, in turn, was negatively correlated with psychological distress. This indirect relationship was confirmed through mediation analysis, revealing that PA’s impact on MH may occur, at least in part, through improved sleep. This finding aligns with previous research showing that regular PA can enhance sleep architecture [119], thereby contributing to better emotional regulation [120] and reduced symptoms of anxiety, depression, and stress [121,122]. From a biopsychosocial perspective, this finding highlights the interdependence of behavioral and psychological processes in shaping well-being. Rather than exerting a direct effect, PA may act as a facilitator within a complex system of interrelated lifestyle habits that collectively support MH [118,123]. This perspective is particularly relevant in non-clinical university populations, where distress levels may be moderate and the benefits of PA may emerge more strongly through behavioral mediators such as sleep. Notably, sleeping fewer than seven hours was associated with more unfavorable outcomes across all psychological variables, underscoring the central role of sleep in MH promotion.

Adherence to the MD also showed a clear and consistent association with psychological well-being in both descriptive and predictive models. Students with greater adherence not only reported higher PA levels and longer sleep duration, but also scored higher in resilience and self-esteem, and lower in psychological distress. These findings are supported by previous studies linking healthy eating with improved emotional regulation, reduced symptoms of depression, stress, and anxiety, and greater resilience [36,124,125,126] Furthermore, this connection may be mediated by factors such as perceived self-care, personal control, and satisfaction derived from maintaining a balanced dietary pattern [127].

Stepwise regression models allowed us to identify the relative weight of each lifestyle habit in predicting psychological well-being, with distinctions by sex. Overall, adherence to the MD emerged as the most robust and consistent correlate, particularly in the models for self-esteem and psychological distress, showing the highest standardized coefficients for both males and females. Its composition—rich in fruits, vegetables, legumes, fish, whole grains, and olive oil, and low in ultra-processed products [36,128]—provides a combination of micronutrients and bioactive compounds with demonstrated anti-inflammatory, neuroprotective, and mood-regulating effects [129,130,131,132]. In contrast, unbalanced dietary patterns high in sugars and fats have been linked to increased emotional vulnerability and a higher risk of affective symptoms [133], underscoring the Mediterranean diet’s value as a central strategy for promoting MH. Among females, adherence to the MD was also the strongest predictor of resilience. Numerous cross-sectional studies have reported a positive association between dietary quality and resilience across populations [124,134,135]. Some authors suggest that individuals with greater resilience are better able to cope with everyday stressors, which, in turn, facilitates the adoption of healthier eating habits [136].

However, this general trend diverged in the case of resilience among males, for whom sleep duration emerged as the most relevant predictor. These findings are consistent with prior research showing that low resilience is associated with heightened sleep reactivity to stress, emotional dysregulation, and difficulty coping with daily challenges—highlighting the role of sleep as a key emotional regulator [56]. This interaction appears particularly significant in males, who have shown greater emotional and cognitive sensitivity to insufficient rest, which may explain their greater dependence on sleep as a protective factor for resilience.

The positive interrelationship between PA, adherence to the MD, and sleep duration suggests a clustering of healthy habits, whose combination appears to exert a synergistic effect on MH [137]. Accordingly, the present findings underscore the importance of developing MH promotion strategies that target multiple lifestyle factors simultaneously, rather than in isolation. Moreover, the results highlight the need to tailor such interventions to sex-specific profiles, as the relative impact of each habit on psychological variables differs between male and female university students.

It is important to note that the participants in this study were educated students who will soon become teachers. This adds substantial value to the findings, as the psychological well-being and lifestyle habits of future educators not only affect their own health but also influence their capacity to serve as role models for their students [137]. Evidence suggests that teachers play a crucial role in promoting healthy behaviors during early childhood—not only through curricular content but also through personal example [138,139]. Therefore, fostering healthy lifestyles among future teachers contributes to the creation of healthier school environments. Integrating structured self-care modules into teacher education programs could prove essential not only for enhancing pre-service teachers’ well-being, but also for empowering them to cultivate a culture of health and emotional resilience in their future classrooms [140].

Beyond their academic and psychological relevance, these findings are also aligned with the broader goals of the 2030 Agenda for Sustainable Development. In particular, they support progress toward SDG 3 (Good Health and Well-being), SDG 4 (Quality Education), and SDG 5 (Gender Equality). Promoting healthy lifestyle habits among future teachers not only enhances their personal well-being but also enables them to act as health-promoting agents in educational contexts. In doing so, they contribute to the development of healthier and more equitable school environments, reinforcing the transformative potential of teacher education in driving systemic improvements in public health and education.

### Limitations, Future Research Directions, and Practical Implications

This study presents several limitations that should be acknowledged. First, its cross-sectional design limits the ability to establish causal relationships between lifestyle habits (PA, MD adherence, and sleep duration) and psychological variables (resilience, self-esteem, and psychological distress). Secondly, all measures were assessed using self-report instruments, which may introduce social desirability bias or inaccuracies in the reported responses. Moreover, the absence of objective measurements of sleep, diet, or PA may reduce the precision and reliability of the data.

Although a negative association between PA and psychological distress was expected, no significant direct relationship was observed. However, the mediation analysis revealed an indirect effect through sleep, suggesting that the MH benefits traditionally attributed to PA may, in non-clinical university populations, operate primarily through its influence on other behaviors—particularly sleep—rather than through a direct psychological mechanism. This finding highlights the need to further explore more complex explanatory models and to conduct longitudinal studies that examine indirect relationships and mediating mechanisms that may modulate the strength and direction of associations between lifestyle and psychological well-being.

Lastly, although the sample was relatively large, participants were recruited from a single university using a convenience sampling method and focused on students in Early Childhood and Primary Education programs. Therefore, the generalizability of the results may be limited. Although the sample represented approximately 27.6% of all students enrolled in the selected degree programs at the University of Granada, it was not randomly selected and cannot be considered representative of the broader student population. This limitation may affect the external validity of the findings and should be taken into account when interpreting the results.

In light of these considerations, future research should prioritize longitudinal designs to assess changes over time and enable the identification of causal pathways. Interventions targeting health promotion should aim to evaluate the cumulative impact of integrated healthy habits on psychological well-being. The inclusion of objective measures—such as accelerometers, actigraphy, or supervised dietary logs—would also help enhance data validity and reduce self-report bias.

Additionally, it is suggested to replicate this study across different academic disciplines and institutional contexts to improve external validity. The use of mixed-method approaches, including qualitative data collection, would offer a richer understanding of the subjective meanings that students themselves attribute to their lifestyle habits and psychological well-being.

Finally, future studies should consider incorporating other mediating or moderating variables, such as motivation toward PA, sleep quality, or emotional eating patterns, as they may help explain some of the paradoxical relationships observed, such as the negative association between PA and self-esteem or the lack of direct association between PA and psychological distress.

In terms of practical implications, universities and teacher training institutions are encouraged to integrate structured self-care modules into their curricula. These should focus on promoting PA, healthy eating, and sleep hygiene as part of the professional development of future educators. Doing so may not only enhance students’ own psychological well-being but also prepare them to act as effective health-promoting role models in their future classrooms. Researchers are also encouraged to evaluate the implementation and long-term impact of such interventions in diverse educational contexts.

## 5. Conclusions

The findings partially supported the initial hypotheses. While no direct association was found between PA and psychological distress, an indirect effect emerged through sleep duration. Adherence to the MD was the most consistently associated lifestyle factor with psychological well-being, particularly for self-esteem and psychological distress, while sleep duration was the strongest association with resilience among men. These results highlight the importance of integrated, sex-sensitive health promotion strategies in university settings and support the inclusion of healthy lifestyle habits as a key component of student well-being programs. Given that participants were future teachers, promoting healthy lifestyle habits in this population is especially relevant, as their well-being directly influences their ability to model and promote health in educational contexts.

## Figures and Tables

**Figure 1 healthcare-13-01197-f001:**
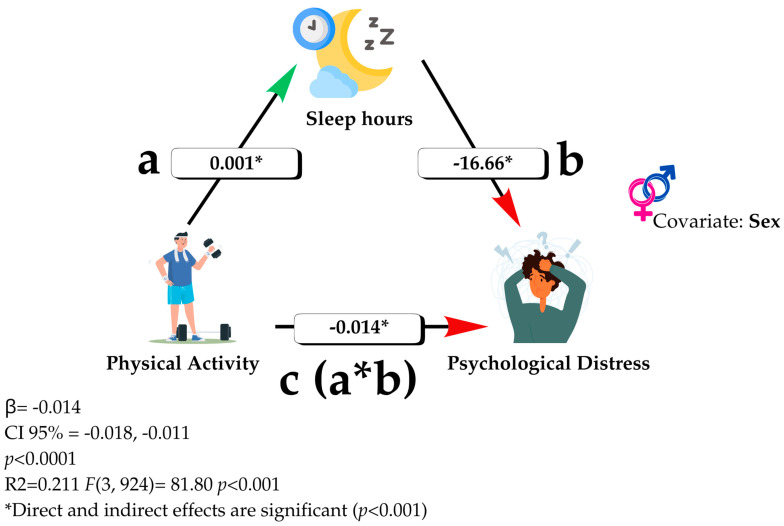
Mediation model of the effect of PA on psychological distress through sleep duration.

**Table 1 healthcare-13-01197-t001:** Differences according to sex.

Variables	Total (n = 928)	Males (n = 303)	Females (n = 928)	Z	*p*-Value
	Mean ± SD	Mean ± SD	Mean ± SD		
PA Combined	4227.12 ± 2281.21	5016.20 ± 2303.98	3844.56 ± 2170.89	−7.283	<0.001
Sleep Hours	7.39 ± 0.66	7.57 ± 0.63	7.30 ± 0.65	−3.676	<0.001
Adherence to MD	6.69 ± 2.23	7.73 ± 1.81	7.29 ± 0.65	−10.299	<0.001
Resilience	64.01 ± 7.61	65.59 ± 6.75	63.24 ± 7.89	−4.664	<0.001
Psychological Distress	45.10 ± 26.76	35.50 ± 22.61	49.75 ± 27.39	−7.507	<0.001
Self-esteem	30.18 ± 5.25	32.31 ± 4.78	29.20 ± 5.19	−8.127	<0.001

Note: PA: Physical Activity; MD: Mediterranean Diet.

**Table 2 healthcare-13-01197-t002:** Differences according to level of physical activity, adherence to Mediterranean diet, and sleep hours.

Variables	Physical Activity			Z	*p*-Value	Post Hoc Test (B)
	LL (n = 29)	ML (n = 304)	HL (n = 595)			
	Mean ± SD	Mean ± SD	Mean ± SD			
Sleep hours	7.14 ± 0.61	7.08 ± 0.52	7.55 ± 0.66	100.30	<0.001 **	a b
Adherence to MD	5.17 ± 2.32	5.77 ± 2.23	7.24 ± 2.03	103.56	<0.001 **	a b
Resilience	62.97 ± 9.38	62.63 ± 8.19	64.77 ± 7.10	14.93	0.001 **	b
Psychological Distress	41.59 ± 30.17	46.90 ± 27.32	44.35 ± 26.29	2.53	0.283	
Self-esteem	29.90 ± 5.25	29.40 ± 5.20	30.60 ± 5.22	9.57	0.008 *	b
	**Mediterranean Diet**			**Z**	***p*-Value**	**Post Hoc Test (B)**
	**LA (n = 83)** **Mean ± SD**	**MA (n = 462)** **Mean ± SD**	**OA (n = 383)** **Mean ± SD**			
PA Combined	2766.11 ± 1837.44	3611.41 ± 2040.50	5284.71 ± 2200.68	148.83	<0.001 **	c d e
Sleep hours	7.09 ± 0.60	7.23 ± 0.61	7.64 ± 0.63	95.63	<0.001 **	d e
Resilience	57.65 ± 10.38	62.65 ± 7.51	67.05 ± 5.41	121.44	<0.001 **	d e
Psychological Distress	71.16 ± 32.54	54.28 ± 23.03	28.29 ± 18.41	291.53	<0.001 **	c d e
Self-esteem	24.64 ± 4.92	28.54 ± 4.45	33.38 ± 4.21	276.30	<0.001 **	c d e
	**Hours of Sleep**			**Z**	***p*-Value**	
	**<7 h (n = 175)** **Mean ± SD**		**≥7 h (n = 753)** **Mean ± SD**			
PA Combined	3415.12 ± 2101.68		4415.82 ± 2281.18	−5.339	<0.001 **	
Adherence to MD	5.76 ± 2.21		6.91 ± 2.17	−6.248	<0.001 **	
Resilience	61.32 ± 7.98		64.64 ± 7.39	−5.678	<0.001 **	
Psychological Distress	60.02 ± 24.89		41.63 ± 25.99	−8.268	<0.001 **	
Self-esteem	27.47 ± 4.68		30.81 ± 5.18	−7.731	<0.001 **	

Note: SD: Standard deviation; Z: Kruskal–Wallis Test; B = Bonferroni test; MD: Mediterranean Diet; PA: Physical Activity; LL: low level of PA; ML: Moderate level of PA; HL: high level of PA; LA: low adherence to MD; MA: Moderate adherence to MD; OA: optimum adherence to MD; a = differences between LL and HL; b = differences between ML and HL; c = differences between LA and MA; d = differences between LA and OA; e = differences between MA and OA; * *p* < 0.05; ** *p* < 0.001.

**Table 3 healthcare-13-01197-t003:** Spearman Correlation Matrix Between Lifestyle Factors, Resilience, Psychological Distress, and Self-Esteem adjusted by Sex.

Variable	Sleep Hours	Adherence to MD	Resilience	Psychological Distress	Self-Esteem
PA Combined	0.327 **	0.445 **	0.182 *	0.018	0.164 **
Sleep hours		0.352 **	0.447 **	−0.417 **	0.506 **
Adherence to MD			0.404 **	−0.601 **	0.616 **
Resilience				−0.316 **	0.485 **
Psychological distress					−0.482 **

Note: PA: Physical Activity; MD: Mediterranean Diet; * *p* < 0.05; ** *p* < 0.001.

**Table 4 healthcare-13-01197-t004:** Model Fit Statistics for Multiple Linear Regression on Resilience, Psychological Distress, and Self-esteem by Sex.

Model	Sex	R^2^	R Adjust	SE	F	*p*-Value
Resilience	Male	0.343	0.337	5.496	52.095	<0.001
	Female	0.216	0.212	7.004	56.971	<0.001
Psychological distress	Male	0.481	0.476	16.363	92.498	<0.001
	Female	0.414	0.412	21.010	146.523	<0.001
Self-esteem	Male	0.524	0.519	3.315	109.626	<0.001
	Female	0.430	0.427	3.927	156.217	<0.001

R^2^ = Coefficient of determination; R adjust = Adjust R^2^; SE = Standard Error; F = F-statistics test.

**Table 5 healthcare-13-01197-t005:** Multiple Linear Regression Results for Resilience, Psychological Distress, and Self-Esteem by Sex.

	B	Standard Error	β	Sig.	95% CI for B		R^2^
					Lower Limit	Upper Limit	
**Model: Resilience (Male)**							0.343
PA Combined	0.000	<0.001	0.133	0.014	−0.001	0.001	
Sleep hours	5.058	0.547	0.475	<0.001	3.982	6.134	
Adherence to MD	1.051	0.207	0.282	<0.001	0.645	1.457	
**Model Resilience (Female)**							0.216
PA Combined	<0.001	<0.001	−0.032	0.406	<0.001	<0.001	
Sleep hours	2.787	0.459	0.229	<0.001	1.886	3.688	
Adherence to MD	1.243	0.137	0.352	<0.001	0.975	1.511	
**Model Psychological distress (Male)**							0.481
PA Combined	0.004	<0.001	0.419	<0.001	0.003	−0.005	
Sleep hours	−10.441	1.629	−0.291	<0.001	−13.647	−7.234	
Adherence to MD	−8.008	0.615	−0.640	<0.001	−9.219	−6.797	
**Model Psychological distress (Female)**							0.414
PA Combined	0.004	<0.001	0.294	<0.001	0.003	0.005	
Sleep hours	−12.279	1.376	−0.291	<0.001	−14.978	−9.574	
Adherence to MD	−6.836	0.410	−0.562	<0.001	−7.640	−6.031	
**Model Self-esteem (Male)**							0.524
PA Combined	<0.001	<0.001	−0.232	<0.001	−0.001	0.001	
Sleep hours	3.181	0.329	0.422	<0.001	2.534	3.834	
Adherence to MD	1.443	0.125	0.547	<0.001	1.197	1.688	
**Model Self-esteem (Female)**							0.430
PA Combined	<0.001	<0.001	−0.186	<0.001	−0.001	0.001	
Sleep hours	2.607	0.55	0.331	<0.001	2.106	3.108	
Adherence to MD	1.236	0.076	0.541	<0.001	1.087	1.386	

Note: CI: Confidence Interval; PA: Physical Activity; MD: Mediterranean Diet.

## Data Availability

The data supporting the reported results are available at reasonable request from the corresponding author.

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
