# Peer review of "Influence of Lifestyle Habits on Psychological Well-Being of University Students: A Quantitative Cross-Sectional Study"

_healthcare, 2025, doi:10.3390/healthcare13101197_

Round 1
Reviewer 1 Report
Comments and Suggestions for Authors
The research is valuable in emphasizing the importance of physical activity among university students.
However, the deficiencies mentioned below need to be remedied.
Introduction
The problem statement of the research should be addressed more clearly because this is very important in terms of highlighting the original value of the research.
Materials and Methods
It is emphasized in the method that the research is cross-sectional. The title can be edited accordingly.
How was the number of participants in the study determined? How was it decided that the number was sufficient? I strongly recommend a power analysis.
What were the criteria for inclusion and exclusion from the study? Should be explained.
What were the validity and reliability indicators of the scales used in the study in your own data? Cronbach alpha values ​​should be calculated.
Results
The findings are beautifully presented.
Discussion
The findings are beautifully presented.
However, some suggestions should be given based on the research results. Suggestions for both practitioners and researchers.
Author Response
REVIEWER 1.
Comment 1: “The problem statement should be addressed more clearly to highlight the original value of the research.”
Response: Thank you for the suggestion. We have revised the final paragraph of the Introduction to clearly articulate the research problem and highlight the originality of jointly analyzing physical activity, Mediterranean diet, and sleep in relation to mental health variables among future teachers (line 97, page 2).
Comment 2: “The title can be edited to reflect the cross-sectional nature of the study.”
Response: The manuscript title has been modified to:
“Influence of Lifestyle Habits on Psychological Well-Being of University Students: A Cross-Sectional Study” (line 1, page 1)
Comment 3: “How was the number of participants in the study determined? How was it decided that the number was sufficient? I strongly recommend a power analysis.
Response: In response to this comment, we conducted a post hoc power analysis using G*Power 3.1.9.7. Assuming an alpha level of 0.05, three predictors (physical activity, Mediterranean diet, and sleep duration), and a medium effect size (f² = 0.15), the analysis revealed that a sample of 928 participants achieves a statistical power of 1.00. This confirms that the sample size was sufficient to detect meaningful associations in the regression analyses. This information has been added to the “Statistical Analysis” section. (line 210, page 5)
Comment 4: “What were the criteria for inclusion and exclusion from the study? Should be explained.”
Response: We have added a specific subsection within “Participants” detailing the inclusion and exclusion criteria (line 123, page 3)
Comment 5: “What were the validity and reliability indicators of the scales used in the study in your own data? Cronbach alpha values should be calculated.”
Response: We have reviewed and updated the manuscript to report Cronbach’s alpha values for all the psychometric instruments used, based on our own data.
Comment 6: “The findings are beautifully presented.”
Response: We appreciate your positive feedback on the clarity of the Results section. Thank you.
Comment 7: “However, some suggestions should be given based on the research results. Suggestions for both practitioners and researchers”.
Response: Thank you for this valuable comment. In response, we have expanded the final part of the Discussion and Section 4.1 ("Limitations, Future Research Directions, and Practical Implications") to include specific and actionable suggestions derived from our findings. (line 536, page 14)
Reviewer 2 Report
Comments and Suggestions for Authors
The manuscript presents important and well-analyzed data examining the association between certain lifestyle habits and psychological well-being among students from one Faculty in Granada, Spain.
I have the following comments:
1: The sampling procedure should be explained.
2: While all students were selected from one faculty, the authors described it as representative (line 107).
3: Some tables split the results by sex, while others did not. This should be reconsidered in the hypothesis. Did the authors intend to assess the sex differences in lifestyle habits and their associations? If yes, all results should be split by sex. If the results did not vary by sex, and sex was shown to have no confounding effect (authors can do formal interaction tests), then it is enough to just adjust the results for sex.
4: The lack of representativeness is a potential limitation.
5: The mediation analysis investigated the contribution of PA and sleep hours to psychological distress, but not self-esteem and resilience. Is there any justification? Did the results vary by sex?
Author Response
The manuscript presents important and well-analyzed data examining the association between certain lifestyle habits and psychological well-being among students from one Faculty in Granada, Spain.
I have the following comments:
Comment 1: The sampling procedure should be explained.
Response: Thank you for your observation. We have expanded the “Procedure” subsection to provide a clearer description of the sampling strategy. (line 194, page 5)
Comment 2: While all students were selected from one faculty, the authors described it as representative (line 107).
Response: Thank you for this important observation. We have revised the manuscript to avoid any overgeneralization regarding representativeness. The sentence in question has been modified to clarify that the sample is representative only within the context of the Faculty of Education at the University of Granada. Specifically, we now state that the sample represents approximately 27.6% of the total number of students enrolled in Early Childhood and Primary Education degrees at this faculty during the 2023/2024 academic year. This clarification is also acknowledged as a limitation in the corresponding section of the manuscript. (line 118, page 3)
Comment 3: Some tables split the results by sex, while others did not. This should be reconsidered in the hypothesis. Did the authors intend to assess the sex differences in lifestyle habits and their associations? If yes, all results should be split by sex. If the results did not vary by sex, and sex was shown to have no confounding effect (authors can do formal interaction tests), then it is enough to just adjust the results for sex.
Response: Thank you for this thoughtful and relevant comment. We confirm that examining sex-based differences was among our secondary research objectives. This has now been explicitly stated in the final paragraph of the Introduction, where we justify the relevance of sex-stratified analyses based on prior literature. Additionally, we have ensured that sex differences are addressed consistently across the Results section. Descriptive and comparative analyses have been stratified by sex where appropriate, and sex was included as a covariate in all regression and correlation models to control for potential confounding effects. While we did not conduct formal interaction tests in the current study, we acknowledge this as a limitation and suggest that future research explore interaction or moderation effects to better understand the role of sex in these associations.
Comment 4: The lack of representativeness is a potential limitation.
Response: Thank you for this important observation. We fully agree, and this limitation is now clearly acknowledged in the “Limitations, Future Research Directions, and Practical Implications” section of the manuscript. Although the sample includes a substantial proportion (27.6%) of students enrolled in Early Childhood and Primary Education programs at the University of Granada, the use of a non-random convenience sample from a single faculty limits the generalizability of the findings. This point has been explicitly stated, and we also recommend that future research replicates the study in other institutions and disciplines to enhance external validity. (line 514, page 13)
Comment 5: The mediation analysis investigated the contribution of PA and sleep hours to psychological distress, but not self-esteem and resilience. Is there any justification? Did the results vary by sex?
Response: Thank you for this thoughtful question. The mediation analysis focused on psychological distress due to both theoretical and empirical grounds. Previous literature has consistently shown that sleep mediates the relationship between physical activity and psychological distress in young adult populations, and our data supported a significant mediation effect in this case. While self-esteem and resilience are key well-being indicators, our models did not explore indirect effects on them due to the absence of strong theoretical support for the mediating role of sleep in those relationships. We have clarified this rationale in the revised manuscript. Regarding sex differences, preliminary analyses did not reveal significant interactions in the mediation paths by sex; however, we recognize this as a limitation and suggest that future studies explore moderated mediation models to examine potential sex-based differences more thoroughly.
Reviewer 3 Report
Comments and Suggestions for Authors
Dear Authors,
Thank you for providing me with the opportunity to review this interesting paper. Below, I have listed my comments:
1) In the introduction, the connection between lifestyle habits and “personal resources” like resilience and self-esteem is implied but not deeply conceptualized. These could be more clearly framed within a theoretical model (e.g., biopsychosocial model).
2) It is mentioned that sex differences will be explored but it is not explained why these might exist based on literature. Brief evidence or rationale supporting this secondary aim would strengthen the case.
3) In the discussion, although causality is mostly handled carefully, some phrases could still imply causation in a cross-sectional study (e.g., predictors could be softened to associations).
4) There are also some parts of underdeveloped discussions. For instance, the absence of a direct PA-distress link is acknowledged but not explored in enough theoretical depth.
I hope this feedback is helpful.
Author Response
REVIEWER 3
Dear Authors,
Thank you for providing me with the opportunity to review this interesting paper. Below, I have listed my comments:
Comment 1: In the introduction, the connection between lifestyle habits and “personal resources” like resilience and self-esteem is implied but not deeply conceptualized. These could be more clearly framed within a theoretical model (e.g., biopsychosocial model).
Response: Thank you for this valuable suggestion. In response, we have revised the Introduction to explicitly incorporate a theoretical framework supporting the relationship between lifestyle habits and psychological variables such as self-esteem and resilience. (line 84, page 2)
Comment 2: It is mentioned that sex differences will be explored but it is not explained why these might exist based on literature. Brief evidence or rationale supporting this secondary aim would strengthen the case.
Response: Thank you for this thoughtful suggestion. We have addressed this point by including a brief but clear justification in the final paragraph of the Introduction. Specifically, we reference prior studies that have documented sex-based differences in both health-related behaviors and psychological outcomes among university students. This addition supports the rationale for exploring sex-stratified associations in our analyses and strengthens the theoretical grounding for this secondary aim. (line 99, page 3)
Comment 3: In the discussion, although causality is mostly handled carefully, some phrases could still imply causation in a cross-sectional study (e.g., predictors could be softened to associations).
Response: Thank you for this important observation. We have carefully reviewed the manuscript to ensure that all language aligns with the cross-sectional nature of the study. Terms that could imply causality, such as “predictors” or “impact,” have been replaced with more appropriate alternatives, such as “associated factors” or “relationships,” where necessary. These revisions ensure that our interpretation of the findings accurately reflects the limitations of the study design and avoids any unintended implication of causal inference.
Comment 4: There are also some parts of underdeveloped discussions. For instance, the absence of a direct PA-distress link is acknowledged but not explored in enough theoretical depth.
Response: Thank you for this valuable comment. In response, we have expanded the relevant section of the Discussion to provide greater theoretical depth regarding the absence of a direct relationship between physical activity and psychological distress.
Reviewer 4 Report
Comments and Suggestions for Authors
Many thanks for the opportunity to review the scientific article titled “Influence of Lifestyle Habits on Psychological Well-Being of University Students: A Quantitative Study.” Please find my comments below:
- In the abstract (line 12), when describing the study group, it gives the impression that the mean age of 21.01 ± 1.95 refers only to the female participants. Please clarify that this age refers to the entire sample, not just the female.
- In the methods section of the abstract, I suggest adding the name of the instrument used.
- In the Introduction (lines 51–53), a citation is missing.
- In Table 1, the row labeled “PA Combined” requires formatting so that the values are consistently presented in one or two lines for better readability.
- In Table 2, I recommend moving the full word “Distress” to the next line instead of splitting it across two lines.
- In the Results section, there are two tables labeled as Table 4. Please update the numbering and center the final table.
- Due to the extensive length of the Discussion section, I suggest dividing it into subsections (e.g., 1. Sex-based differences; 2. Influence of lifestyle habits on key psychological well-being variables).
- The authors note that one of the study’s limitations is the use of self-report questionnaires, which are subjective and may lead to bias or inaccuracies in responses. Does this limitation not compromise the reliability of the results? I kindly ask the authors to address this issue.
- In the References section, there is an error in citation number 42.
Author Response
Comment 1: In the abstract (line 12), when describing the study group, it gives the impression that the mean age of 21.01 ± 1.95 refers only to the female participants. Please clarify that this age refers to the entire sample, not just the female.
Response: Thank you for pointing this out. We have clarified in the abstract that the reported mean age (21.01 ± 1.95) refers to the total sample, not just female participants. (line 12, page 1)
Comment 2: In the methods section of the abstract, I suggest adding the name of the instrument used.
Response: Thank you for these two useful suggestions. We have revised the Methods section of the abstract to include the full names of all validated instruments used in the study. (lines 14-16, page 1)
Comment 3: In the Introduction (lines 51–53), a citation is missing.
Response: Thank you for your observation. The authors believe that the statement in question is conceptually supported and further developed throughout the rest of the Introduction, where relevant empirical evidence is cited in detail. Therefore, we have not added an additional citation at that specific point, as it functions as a general linking statement that is elaborated on and substantiated in the subsequent paragraphs.
Comment 4: In Table 1, the row labeled “PA Combined” requires formatting so that the values are consistently presented in one or two lines for better readability.
Response: Thank you for this observation. We have revised the formatting of the “PA Combined” row in Table 1
Comment 5: In Table 2, I recommend moving the full word “Distress” to the next line instead of splitting it across two lines.
Response: Thank you for your attention to detail. The formatting of Table 2 has been corrected so that the word “Distress” now appears entirely on a single line.
Comment 6: In the Results section, there are two tables labeled as Table 4. Please update the numbering and center the final table.
Response: Thank you for identifying this oversight. We have corrected the table numbering so that each table has a unique and sequential label.
Comment 7: Due to the extensive length of the Discussion section, I suggest dividing it into subsections (e.g., 1. Sex-based differences; 2. Influence of lifestyle habits on key psychological well-being variables).
Response: Thank you for this thoughtful recommendation. After carefully considering the structure of the Discussion section—especially following revisions prompted by the comments of previous reviewers—the authors believe that the current version maintains a coherent and logically ordered narrative. The section flows progressively from general findings to specific aspects, including sex-based differences, associations with lifestyle habits, mediation effects, and implications. Therefore, we respectfully prefer to retain the current format, as it aligns with the intended style and structure of the manuscript.
Comment 8: The authors note that one of the study’s limitations is the use of self-report questionnaires, which are subjective and may lead to bias or inaccuracies in responses. Does this limitation not compromise the reliability of the results? I kindly ask the authors to address this issue.
Response: Thank you for raising this important point. We agree that the use of self-report instruments introduces the possibility of response bias, such as social desirability or recall inaccuracies. However, we have used widely validated and reliable questionnaires (e.g., IPAQ-SF, KIDMED, CD-RISC, DASS-21, and Rosenberg Self-Esteem Scale), all of which have demonstrated strong psychometric properties in previous research and in the present sample. Furthermore, we acknowledge this limitation explicitly in the manuscript and recommend that future research incorporate objective measures (e.g., accelerometers, actigraphy, or food diaries) to complement self-reported data. While self-report methods may affect the precision of certain responses, we believe they do not compromise the overall reliability or integrity of the study’s findings.
Comment 9: In the References section, there is an error in citation number 42.
Response: Thank you for pointing this out. We have reviewed and corrected reference number 42 in accordance with the journal’s formatting guidelines. (line 700, page 17)
Round 2
Reviewer 1 Report
Comments and Suggestions for Authors
Thank you for your revisions.
Reviewer 2 Report
Comments and Suggestions for Authors
No more comments.
Reviewer 3 Report
Comments and Suggestions for Authors
Thank you, authors, for revising the manuscript and good luck with the rest of the process.